



# Measurements of atmospheric radioactivity dose rates over the North Pacific after the Fukushima Daiichi nuclear power plant accident during the period March 2011 - March 2015

Kuo-Ying Wang[1], Philippe Nedelec[2], Hannah Clark[2], and Neil Harris[3]

[1]Department of Atmospheric Sciences, National Central University, Chung-Li, Taiwan
[2]Laboratoire d'Aérologie, Centre National de la Recherche Scientifique, Observatoire Midi-Pyrénées, 14 Avenue E. Belin, 31400 Toulouse, France
[3]Centre for Environment and Agricultural Informatics, Cranfield University, Cranfield, UK

**Correspondence:** Kuo-Ying Wang (kuoying@mail.atm.ncu.edu.tw)

**Abstract.**

On 11 April 2011, a magnitude 9.0 earthquake occurred about 154 km northeast of the Fukushima Daiichi Nuclear Power Plant (FNPP1; $37.420°N$ and $141.033°E$). Here we present continuous measurements of the atmospheric dose rates after the Fukushima accident over North Pacific atmosphere by a fleet of thirteen in-service global container cargo ships and another

at a Tokyo port site from the Taiwanese PGGM (Pacific Greenhouse Gases Measurements) project. The continuous measurements of atmospheric dose rates were collected from a total of 294 cruises with 41,485 measurements during the period March 2011 to 2015. In this work, we identify three key aspects of the impacts of the radioactive materials following the Fukushima accident: The altitude effect, the land surface effect, and the transported effect. We showed measurements of air dose rates over the land surface areas and over the oceanic atmosphere. The striking differences in the air dose rates measured during

the 2011-2015 period clearly identify the deposition effects and the effects associated with radioactive materials transported from the FNPP1. Air dose rates measured over the land surface areas were resulted from a combination of the effects from the deposition of radioactive materials on the surface and the radioactive materials contained in the airborne particles. Air dose rates measured over the oceanic atmosphere contains radioactive effects from airborne particles. The data of air dose rates over the North Pacific atmosphere show the eastward transport of radioactivity. Eastward transport of radioactive ma-

terials had been observed after 11 Mar 2011. Monitoring data show that the export of radioactive materials to the Pacific atmosphere occurred after March 2011, and in 2012, 2013, 2014, and 2015. All measurment raw data reported in this work are available at https://doi.org/10.6084/m9.figshare.9757697 (Wang et al., 2019), and the calibrated data files are available at http://doi.org/10.5281/zenodo.3468896 (Wang, 2019). Our data can help to further develop and verify model for the atmospheric dispersion of nuclear materials from the FNPP1 over the land surface areas and over the North Pacific atmosphere.





## 1   Introduction

Safety of nuclear power plants is a big issue and a significant health concern for the general public (Ten Hoeve and Jacobson, 2012; Brumfiel, 2013). There are 33 serious accidents associated with nuclear power plants worldwide during 60 years of 1952-2011 (Rogers, 2011). This statistic accounts for about one serious accident per 22 months. While the existence of a
nuclear power plant is a voluntary risk to the host country, the transport of the contaminated particles in the air comprises an ultimate involuntary risk for the neighboring countries and the global environment. In the context of this occurrence rate, it is necessary to systematically and continuously collect data of atmospheric nuclei concentrations for risk assessment and management associated with the nuclear power plants.

The main character of a nuclear accident is that the source of a nuclear accident is highly localized, but the effect ripples
around the globe. For example, the Chernobyl nuclear accident occurred on 26 April 1986. Ground-level monitoring data revealed elevated levels of atmospheric nuclei concentrations over northern Europe on 26 April 1986. Model simulations showed high levels of radiative nuclei from the Chernobyl nuclear power plant accident were transported across Europe by the atmospheric winds (Quélo et al., 2007; Evangeliou et al., 2016).

On 11 April 2011, a magnitude 9.0 earthquake occurred at a distance about 154 km northeast of the FNPP1 (Korsakissoki
et al., 2013). The earthquake produced a 13-meter height tsunami at Fukushima area, inundated the power plant with seawater, knocked off electricity to the primary water cooling pumps, lost 12 out of 13 emergency diesel generators, resulting in the overheating and explosions of the boiling-water reactors and releases of nuclear radioactive materials into the atmosphere (Lipscy et al., 2013). The collapse of the Fukushima Daiichi nuclear power plant marked another wake-up call for accessing the risk associated with the operation of nuclear power plants (Rose and Sweeting, 2016).

In this work, we report the continuous monitoring of atmospheric radioactivity dose rates over the Pacific regions from March 2011 to 2015. The monitoring data helps us to understand the spatial and temporal scale of dispersion of radioactive nuclei from the Fukushima Daiichi nuclear power plant. The data collected and reported in this work are openly and openly available (in the supplementary material of this work). We hope that the data will aid the validation of atmospheric models for modeling atmospheric dispersions of radioactive materials from the nuclear power plant. We conclude by noting that, in
the context of public health and welfare, continuous monitoring and modeling works are necessary to ensure the risk of using nuclear power plants are well monitored.

## 2   Data and Methods

### 2.1   The PGGM Monitoring Platform Over the North Pacific

As soon as the news of the Fukushima accident was shown on the TV news, we expected that the subsequent atmospheric
transport and dispersion of contaminated nuclei from the Fukushima Daiichi power plant would occur over the North Pacific regions. What had happened over Europe after the 26 April 1986 from the Chernobyl nuclear power plant (Evangeliou et al., 2016) could have taken place over the Pacific areas. The Pacific Greenhouse Gases Measurement (PGGM) project has equipped





**Table 1.** List of Instruments, Measurement Sites, and Observational Period.

| Number | Instrument Type | Serial Number | Measurement Sites | Observational Period |
|--------|-----------------|---------------|-------------------|----------------------|
| G01 | B-20 | 01059 | Tokyo | $03/2011 - 09/2015$ |
| G02 | B-20 | 01009 | Container Ships | $04/2011 - 12/2014$ |
| G03 | G-10 | 02113 | Container Ships | $03/2011 - 05/2015$ |
| G04 | G-10 | 02111 | Container Ships | $04/2011 - 03/2015$ |
| G05 | G-10 | 02120 | Container Ships | $03/2011 - 06/2015$ |
| G06 | G-10 | 02118 | Container Ships | $03/2011 - 04/2015$ |
| G07 | G-10 | 02115 | Container Ships | $04/2011 - 03/2015$ |
| G08 | G-10 | 02114 | Container Ships | $04/2011 - 12/2014$ |
| G09 | G-10 | 02119 | Container Ships | $04/2011 - 01/2015$ |
| G010 | B-20 | 00993 | Container Ships | $04/2011 - 01/2015$ |
| G011 | B-20 | 01060 | Container Ships | $04/2011 - 03/2015$ |
| G012 | B-20 | 01054 | Container Ships | $04/2011 - 04/2015$ |
| G013 | B-20 | 01055 | Container Ships | $04/2011 - 08/2014$ |
| G014 | B-20 | 01053 | Container Ships | $04/2011 - 03/2015$ |

a fleet of nine global container cargo ships from Evergreen Marine Corporation (EMC) with carbon dioxide ($CO_2$) analyzers to collect CO2 data for climate research since 2009 (Wang et al., 2011a; Wang et al., 2011b).

This good long-term collaboration with the EMC enables us to quickly conceive a project on monitoring atmospheric radioactivity dose rates over the North Pacific regions right after 11 March 2011. The quick response and flexibility associated with a commercial company make possible the urgent deployment of monitoring devices for radioactivity dose rates.

### 2.2    The Monitoring Devices and Calibrations

The radioactivity dose meters from Thermo Scientific RadEye B20 and RadEye G10 were used in this work. The measuring
range of gamma dose rate for the B20 sensor is 0-2 mSv/h (Stenstad et al., 2014; Poulsen et al., 2014; Horsburgh and Higgins, 2016). The measuring range for the G10 sensor is 0.05 $\mu$Sv - 100 mSv/h (Jibiri and Obarhua, 2013; Sahoo et al., 2016). Both B20 and G10 sensors use an energy compensated GM-tube detector. Table 1 shows a list of instrument type, serial number, location, observational period, the lowest calibration values (10 $\mu$Sv/h), calibrated results, and calibration dates for all fourteen sensors used in this work. The first calibrations occurred during March-April 2011 before the sensors were deployed to a land-
based site and container ships. The second calibrations took place on 6 November 2015 when all the sensors were discharged from the container ships and land-based site and returned to National Central University to prepare for calibrations.





**Table 2.** List of Instruments, Calibration Values for 10 $\mu$Sv Does Rates, and Calibration Dates.

| Number | Instrument Type | Cal-1 | Cal-2 | Cal-1 Date | Cal-2 Date |
|--------|-----------------|-------|-------|------------|------------|
| G01  | B-20 | 9.55  | 10.14 | 01/04/2011 | 06/11/2015 |
| G02  | B-20 | 9.55  | 10.12 | 17/03/2011 | 06/11/2015 |
| G03  | G-10 | —     | 10.36 | —          | 06/11/2015 |
| G04  | G-10 | 10.56 | 10.64 | 24/03/2011 | 06/11/2015 |
| G05  | G-10 | 9.99  | 10.30 | 24/03/2011 | 06/11/2015 |
| G06  | G-10 | 9.97  | 9.96  | 24/03/2011 | 06/11/2015 |
| G07  | G-10 | 10.14 | 10.10 | 24/03/2011 | 06/11/2015 |
| G08  | G-10 | 10.01 | 10.60 | 24/03/2011 | 06/11/2015 |
| G09  | G-10 | 10.12 | 9.87  | 24/03/2011 | 06/11/2015 |
| G010 | B-20 | 10.00 | 10.24 | 17/03/2011 | 06/11/2015 |
| G011 | B-20 | 8.86  | 9.36  | 08/04/2011 | 06/11/2015 |
| G012 | B-20 | 9.56  | 9.80  | 08/04/2011 | 06/11/2015 |
| G013 | B-20 | 9.92  | 10.34 | 08/04/2011 | 06/11/2015 |
| G014 | B-20 | 9.05  | 9.94  | 08/04/2011 | 06/11/2015 |

The calibrations of all sensors were measured against the PTW 1 liter spherical ionization chamber type 32002 (PTW TM32002, SN: 298; PTW-Freiburg, Freiburg, Germany) radiation detector (Schuller et al., 2015) at the Nuclear Science and Technology Development Center at National Tsing Hua University.

The 137-Cs provided the radioactive source for sensor calibrations, and at the strength of 11 GBq, 18.5 GBq, and 1850 MBq (1 July 1996). Calibrations were conducted at four designated dose rates for each sensor: 10 $\mu$Sv/h, 80 $\mu$Sv/h, 200 $\mu$SV/h, and 800 $\mu$SV/h. The calibration results for 10 $\mu$Sv does rates are shown in Table 2. Detailed calibration results against four designated dose rates for each sensor are shown in Fig. 1. The calibration results were analyzed with a linear regression model (Press et al., 1992; Wang and Chau, 2013), which compares designated (given) dose rates with the measured dose rates for

each sensor. The analysis shows that the calibration factors, determined as the ratios of measured dose rates to that of the calibrated dose rates, of each B20 and G10 sensor are consistently within 90designated dose rates for dose rates lower than 100 $\mu$Sv. Large deviations exist for air doses higher than 100 $\mu$Sv, with the largest discrepancies occur at 800 $\mu$Sv/h. Some sensors perform remarkably well (within 5(packages G03, G05, G06, and G07). These are all G10 sensors.

     The calibration factors are analyzed with a linear regression model (Press et al., 1992; Wang and Chau, 2013) between the

calibration factors and the measured dose rates. The analysis results for each sensor are shown in Fig. 1. Since the measured dose rates on container ships and a land-based site are less than 0.50 $\mu$Sv/h, the fitted equation $y$ of calibration factors as a function of measured dose rates $x$, can be approximated by a constant factor A when $x$, the measured dose rates, is close to zero. The factors A for the first calibrations in 2011, A1, and the second calibrations in 2015, A2, were calculated for each



**Figure 1.** Calibration of fourteen sensors for measurements of air dose rate. Each panel comprises of five sub-panel. Upper left shows the first comprison between the designated dose rates (Des Span) vesus measured dose rates (Obs Span); upper right shows calibration factors with respect to the designated dose rates. Lower two sub-panels are results from the second calibrations. The lowest sub-panel shows a linear model of calibration factors with respect to days.





sensor. We then build a lookup table of calibration factors, Ai, for each day of measurements by linearly interpolating A1 and
A2 in time. This look-up table of Ai is then used in the calibrations of measured dose rates (Wang et al., 2019) for each B20
and G10 sensors (Wang, 2019). The results and distribution of Ai for each sensor are shown in Fig. 1.

### 2.3 The Monitoring Sites

The sensor number G01 measured the radioactivity dose rates at a land-based site located at the Tokyo Port (Fig. 2). The
measurements were taken at three locations: The attic of the office building ($35.615°N$, $139.781°E$), the A 4 terminal gate
($35.615°N$, $139.781°E$), and the calling ships ($35.612°N$, $139.778°E$). The Tokyo port is located at a distance of 228 km
and southwest to the FNPP1. The rest of thirteen sensors were deployed onboard the container ships operated over the North
Pacific (Fig. 3). Table 1 shows a list of air dose sensors and measurement period.

### 2.4 Data Collection and Validation

Measurements were taken 6-hourly onboard the ship. These measurements were made manually by officers of the ship at the
compass deck, which is located at the top of the ship. The readings were taken from 6 directions: east, west, north, south,
upward, and downward directions. Measurements at Tokyo Port were also manually taken at 00 UT and 06 UT hours during
the working day. The readings were also taken at six directions as those onboard the ships. In order to test the importance of
sensitivity of measurements as a function of distance from the ground, two additional measurements were taken: at 50 cm and
10 cm above the ground at the attic of the office (Wang et al., 2019).

## 3 Results

### 3.1 Radioactive Dose Rates over the Land Surface in the Tokyo Port

Fig. 4 shows time-series measurements of radioactive dose rates at the attic of Tokyo Port office from March 2011 to September
2015. The green dots show all measurement data. Vertical red squares show monthly mean data (red crosses), and the 25th
(lower), the 50th (middle), and the 75th (top) percentiles of the data. Vertical lines indicate the maximum and minimum values
of measurements within a month.

In the first two months after 11 March 2011, the highest monthly radioactive dose rates measured at Tokyo Port office were
at 0.40 and 0.45 $\mu$Sv/h. The highest monthly dose rates gradually drop with time. It was 0.18 $\mu$Sv/h in March 2012, one year
after the disaster. Two years after the disaster, the measurements were at 0.15 $\mu$Sv/h in March 2013; at about 0.14-0.15 $\mu$SV/h
in the third year after the disaster (2014); 0.13-0.14 $\mu$SV/h in the fourth year (2015), and 0.12-0.14 $\mu$SV in the fifth year. Given
that the overall reduction trends in radioactive dose rates elevated high dose rates still occur occasionally in August 2012, in
August 2013, in November 2014, and in September 2015.

The highest monthly mean radioactive dose rates measured at 0.17 $\mu$Sv/h on March 2011. It reduced to 0.12 $\mu$Sv/h on March
2012 but went up to higher than 0.12 $\mu$Sv in the twelve months before March 2013. The dose rates then went down again to

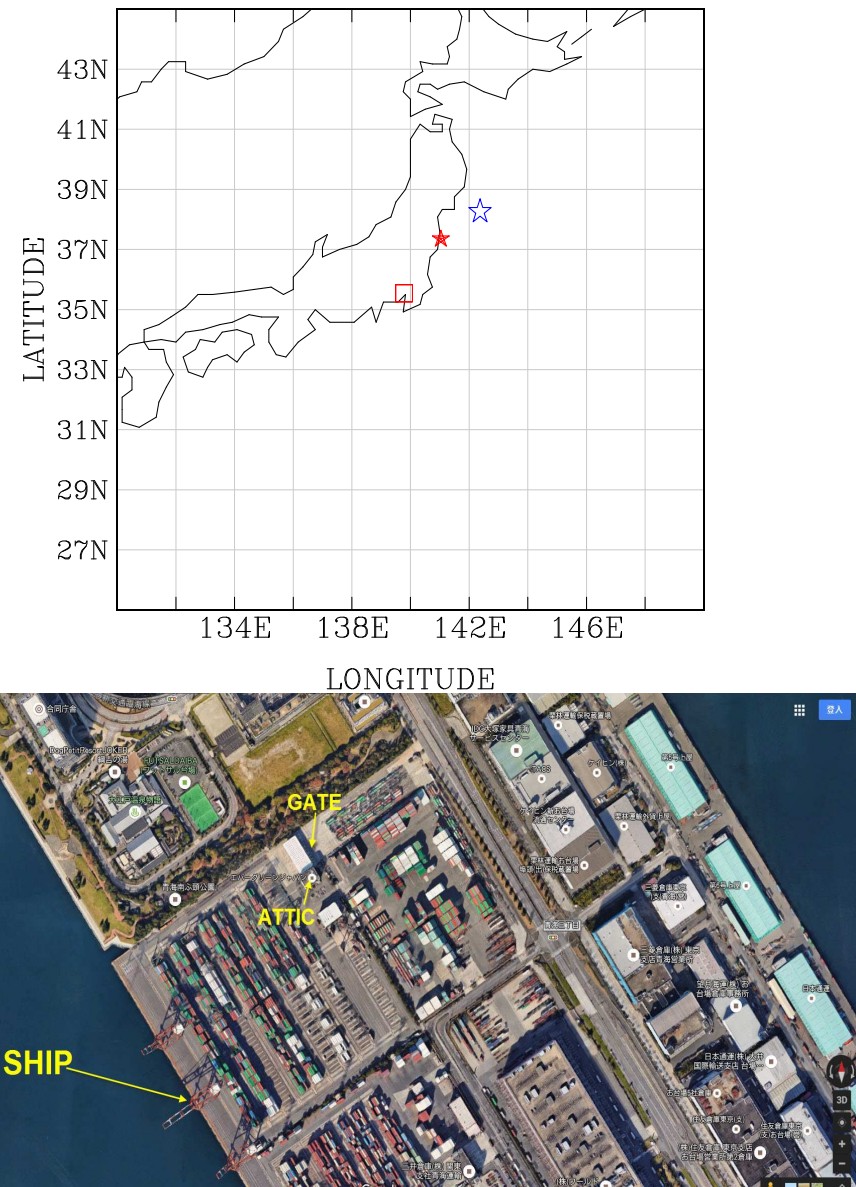

**Figure 2.** Sites of air dose measurements. (Upper panel) Locations for the measurements of radioactivity dose rates at Tokyo Port site (open red square), 11-March-2011 earthquake epicenter ($142.369° E, 38.322° N$; open blue star), and FNPP1 ($141.033° E, 37.423° N$; solid red star). (Lower panel) A zoom-in satellite image (© Google Maps) provided by the EMC Tokyo office, showing an attic area of the Tokyo Port office (ATTIC; $139.781° E, 35.615° N$), A4 gate (GATE; $139.781° E, 35.615° N$), and location for calling ships (SHIP; $139.778° E, 35.612° N$).

slightly higher than 0.10 $\mu$Sv/h in the twelve months before March 2014. The measurements went down to close or lower



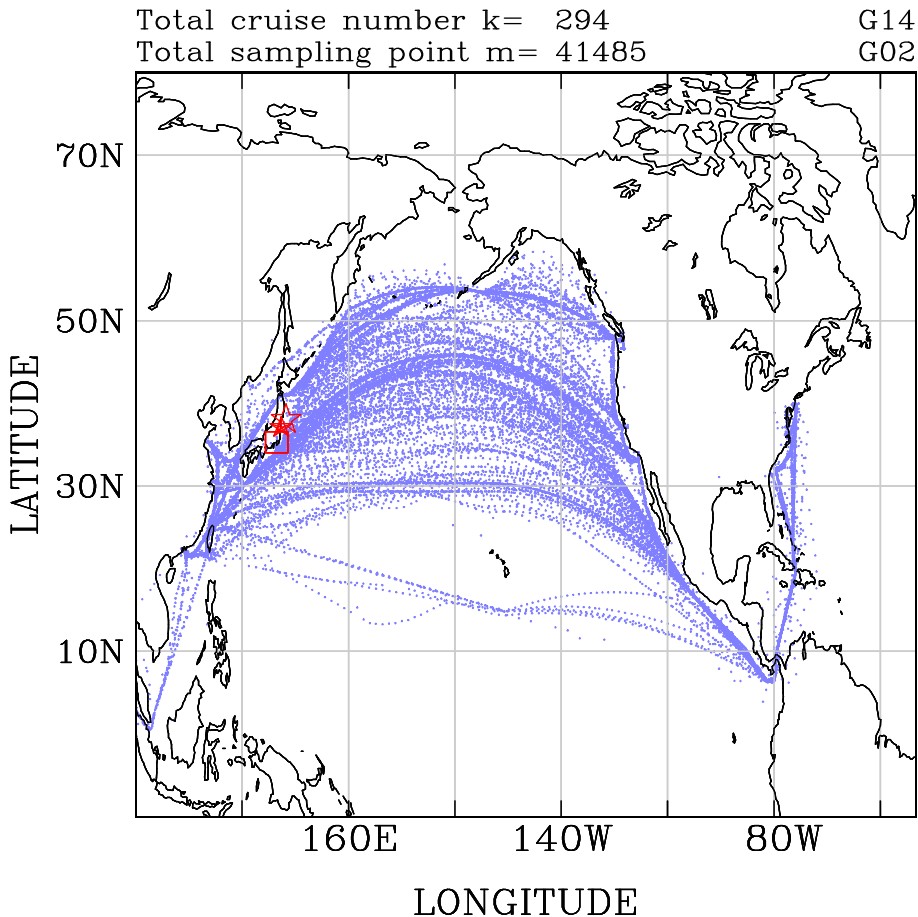

**Figure 3.** Measurements of radioactivity dose rates over the North Pacific and the Northweste Atlantic marine atmosphere. Open star is the location of the 11-March-2011 earthquake epicentre, solid star is the FNPP1 site, and open squre is the measurements at sites at the Tokyo port office area.

115    than 0.10 $\mu$Sv/h in the twelve months before March 2015. These data exhibit the quick impact of radioactive materials from Fukushima after the disaster had occurred, and the subsequent reductions in dose rates with time. After five and a half years, the radioactive dose rates measured at an office attic at Tokyo Port still keeps dropping. These data reveal the direct impact of the radioactive materials transported from the Fukushima Daiichi nuclear power in Tokyo area.

Fig. 5 shows time-series measurements of radioactive dose rates at the A4 gage. The notable increase in radioactive dose

120    rates occurred in the first year after the 11 March 2011 disaster. The continuous reduction in the radioactive dose rates from March 2011 to September 2015. The occurrence of short-period elevated dose rates appeared in subsequent years.

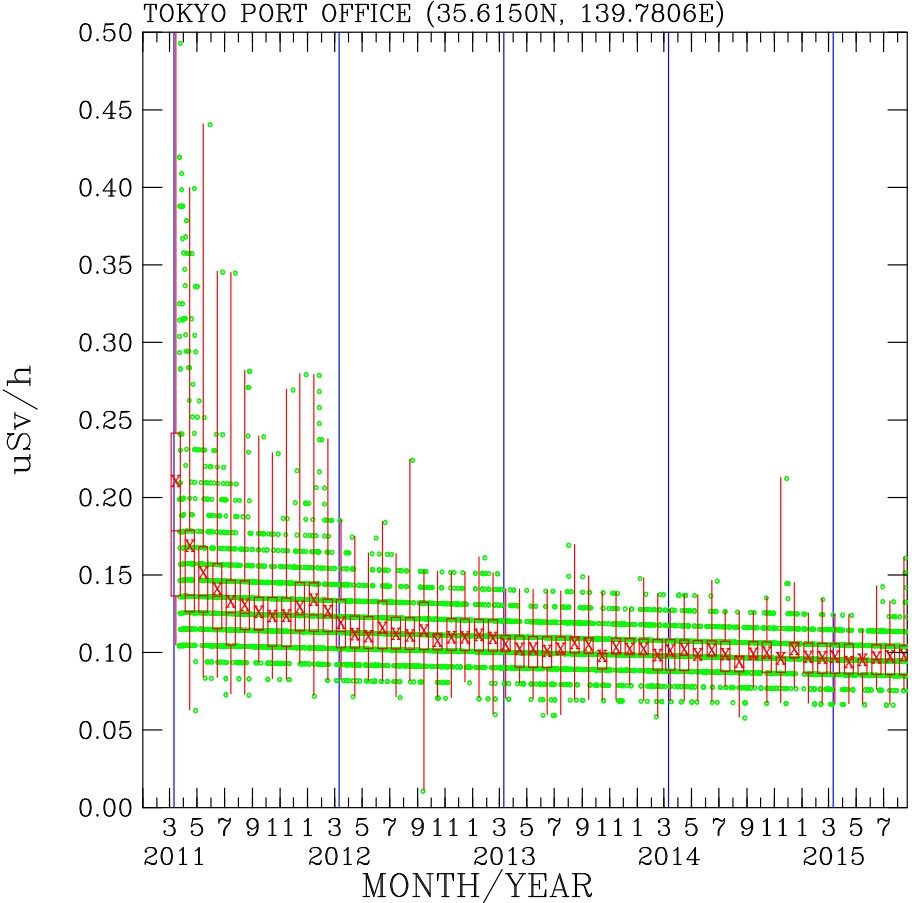

**Figure 4.** Radioactivity dose rates measured made at a Tokyo site. Green dots indicate all data; red box indicate the 25th, 50th and 75th percentiles of all data in a month. The vertical line indicates the maximum and minimum air dose rates.

### 3.2 Radioactive Dose Rates Over the Water Surface in the Tokyo Port

Fig. 6 shows time-series measurements of radioactive dose rates on the calling ships in the Tokyo Port. In the first year after 11 March 2011, 10 out of 12 months of the monthly mean dose rates are above 0.07 $\mu$Sv/h. In the second year (2012), 4 out of 12 months of the monthly mean dose rates are above 0.07 $\mu$Sv/h. In the third year (2013), 7 out of 12 months of the monthly mean dose rates are higher than 0.07 $\mu$Sv/h. In the fourth year (2014), 2 out of 12 months of the monthly mean dose rates are higher than 0.07 $\mu$Sv/h. In the first 9 months of the fifth year (2015), all monthly mean dose rates are lower than 0.07 $\mu$Sv/h. Hence, the measured dose rates are showing reduction trends as well.

Fig. 7 compares the measurements at the two land-site locations in the Tokyo Port: attic of the office building and the A4 gate. Both measurements show consistently distinctive and elevated levels of radioactive dose rates in the twelve months after 11 March 2011. High levels of dose rates graduate decrease with time. However, distinctive discrepancies between the

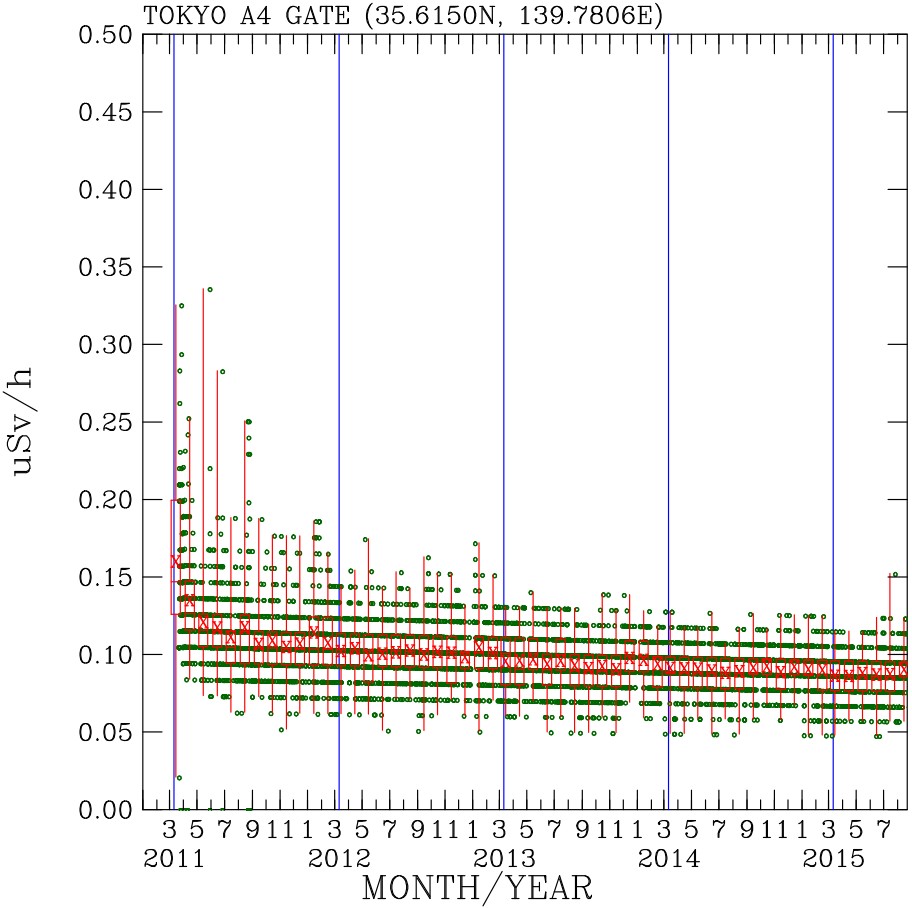

**Figure 5.** Radioactivity dose rates measured at the Tokyo Port A4 gate.

measurements at the attic of the office and the A4 gate. The measurements at the attic office were higher than the measurements made at the A4. The discrepancies between these two data set are found in the time close to 11 March 2011 and become close to each other at the time of September 2015. The main reason for these discrepancies is due to the altitudes of measurements taken in the attic and the A4 gate. Two additional measurements were taken in the attic of the office. Elevated dose rates from the attic measurements than at the A4 gate indicate the presence of deposited radioactive materials on the ground. Highest radioactivity levels were measured at 10-cm altitude from the ground; followed by 50-cm and 1-m from the ground. Elevated results were measured close to the surface, indicating the effect of radioactivity from deposited radioactive materials on the surface.

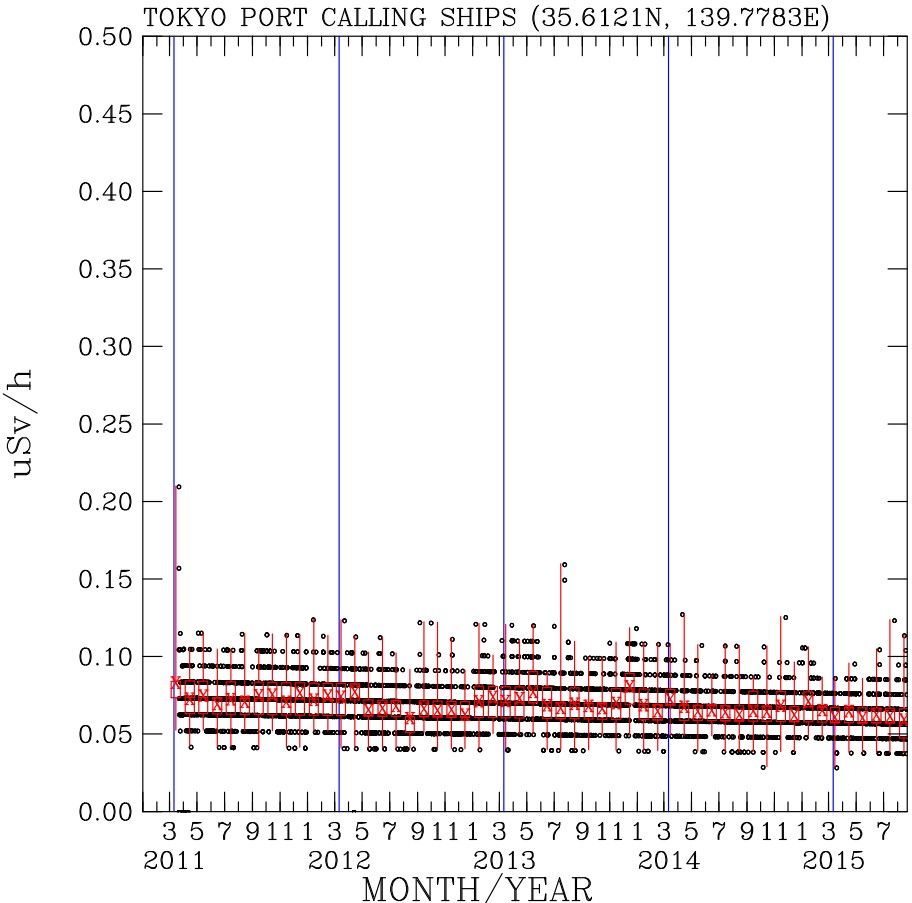

**Figure 6.** Radioactivity dose rates measured on the calling ships at the Tokyo Port.

### 3.3 Effect of the Ground Level Deposition of Radiative Materials: The Tokyo Port Calling Ships

Fig. 8 compares measurement made at the attic of the office to those made on the compass deck of the calling ships at the Tokyo Port. The attic measurements are clearly and distinctively higher than the measurements made at the calling ships. The measurements at the Tokyo Port were made over the land surface where deposition of radioactive materials accumulated. The measurements on the calling ships at the Tokyo Port were made over the ocean, where deposited radioactive materials were sunk into the ocean water. These comparisons again demonstrate the importance of radioactive materials deposited on the ground to provide the largest source of measured radioactive dose rates. In September 2015, 54 months after 11 March 2012, the monthly mean radioactive dose rates over the land-based site at the attic of the office building was 0.10 $\mu$Sv/h, compared with 0.06 $\mu$Sv/h for measurements on the calling ships over the ocean water in the Tokyo Port. The presence of radioactive materials over the land still exert the effect on radioactive dose rates when compared with those over the ocean water.

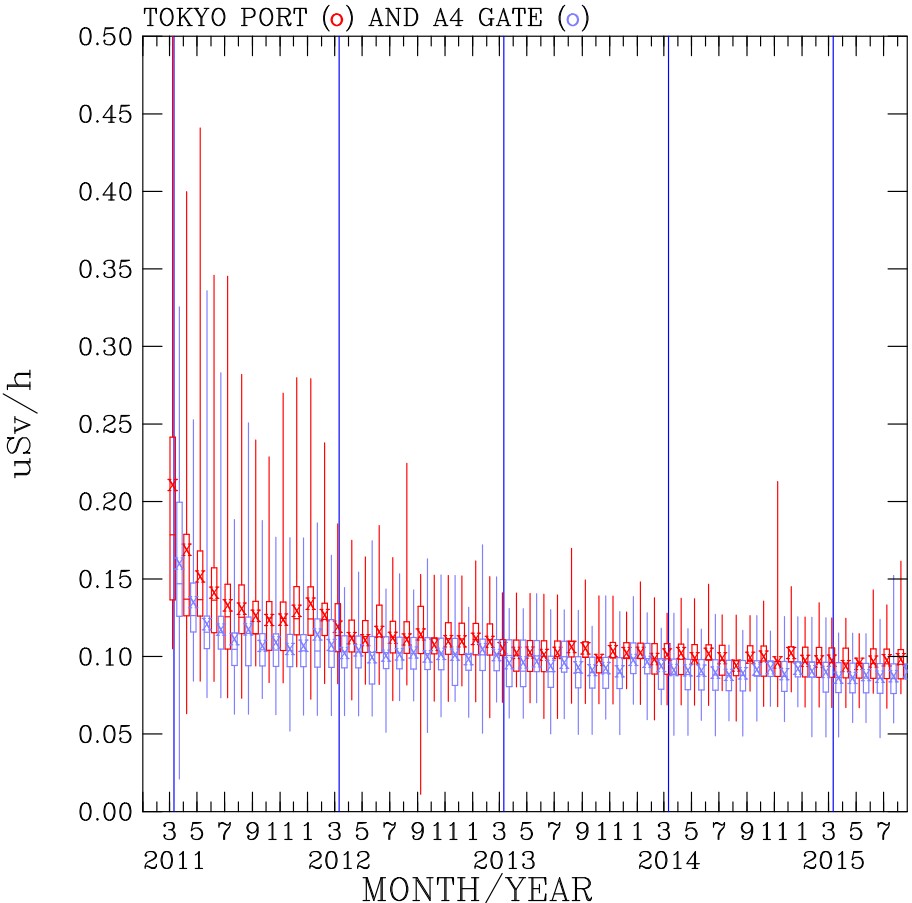

**Figure 7.** Comparisons of radioactivty dose rates measured at the Tokyo Port office attic (red colored) with those measured at A4 gate (blue colored).

The differences between the land surface measurements and the measurements over the ocean surface vary with time. These differences are smaller during January-June and August 2013; during November 2013 to January 2014; during March, September, and November in 2014; and during January 2015. The above periods show elevated dose rates measured over the water surface on calling ships, leading to smaller differences in dose rates between the land-based and ship-based measurements. Most of these variabilities had occurred during the autumn-winter-spring months when Tokyo Port was downwind from the inland area. As such, the calling ships berthed at the Tokyo Port intercepted radiative dust materials transported from land toward the ocean, giving rise to the radioactive dose measurements over calling ships.

## 3.4 Effect of the Ground Level Deposition of Radioactive Materials: The Pacific Sailing Ships

In order to further compare measurements made over the land surface to the measurements made over the ocean water surface, Fig. 9 compares measurements made at the attic of the office building over the land surface to the measurements made on the

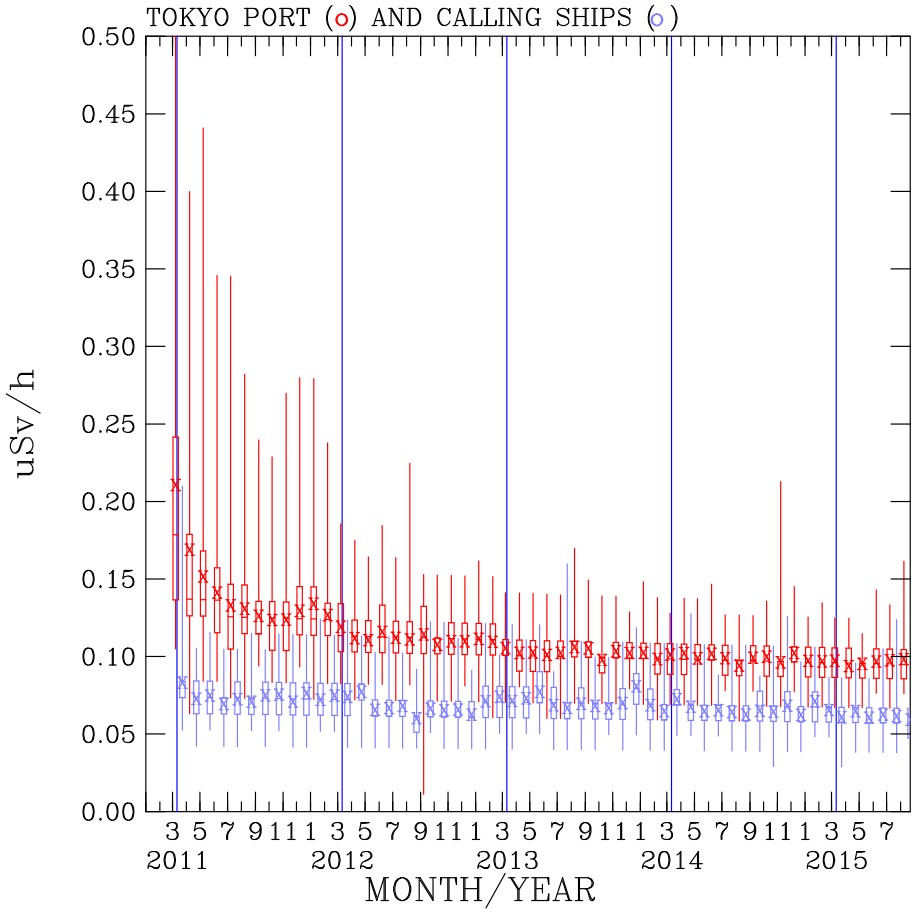

**Figure 8.** Comparison of radioactivity dose rates measured at the Tokyo Port office attic (red colored) with those measured on the calling ships at the Tokyo Port.

compass deck of the Pacific sailing ships. The elevated dose rates occurred in the few months after 11 March 2012 were shown on the Pacific sailing ships (more results are shown in Section 3.6). The measurements over the land surface at Tokyo Port were consistently higher than the measurements on the Pacific calling ships. However, the differences between the land surface measurements and the Pacific measurements vary with time. Both measurements were more close to each other after 11 March 2013 than before 11 March 2013.

Larger differences between the land surface and the Pacific measurements occur in time close to 11 March 2011, indicating the effect of deposited ground-level radioactive materials. The gradual convergence of air dose rates after 11 March 2013, and the variabilities of this closeness of the measurements over the land surface and the Pacific Ocean indicate waves of radioactive materials been transported from land areas to the North Pacific atmosphere.

    The differences between the land-surface measurements at Tokyo Port and the Pacific sailing ships (Fig 9) are smaller than

the differences between the land-surface measurements and the ocean-surface measurements on the calling ships at Tokyo Port

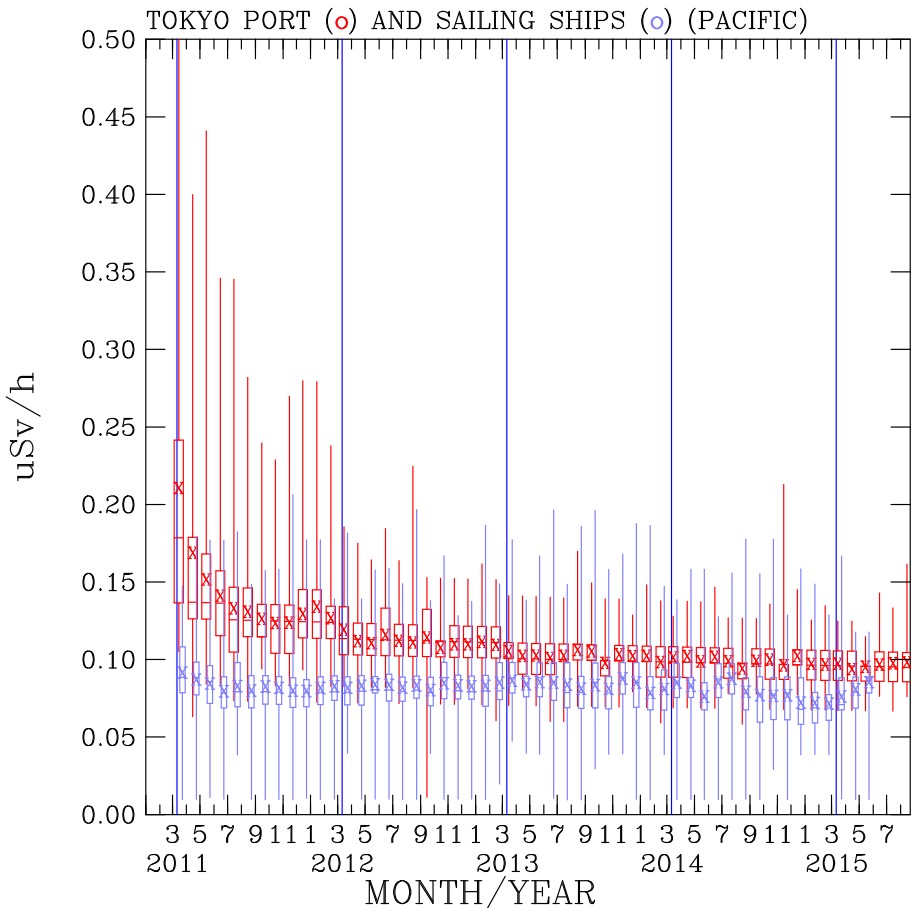

**Figure 9.** Comparison of radioactivity dose rates measured made at the Tokyo Port (red colored) with those measured on the Pacific saling ships (blue colored).

(Fig 8). The calling ships at the Tokyo Port include the container ships traveling over the Pacific and Indian Oceans. The winds at the Tokyo Port can also be impacted by the oceanic winds from low latitudes with low air dose rates.

On the other hand, the Pacific sailing ships traveled mostly at latitudes north of 45 degrees North, which is predominantly downwind of Japan land area. Hence the Pacific sailing ships can be impacted by air plumes carrying radioactive materials from the land areas downwind to the marine environment. Interception of the radioactive materials by the Pacific sailing ships increased dose rates measured onboard the compass desk.

The measurements onboard the Pacific sailing ships are especially close to the land-based measurements during the autumn-winter-spring months when the North Pacific is located downwind of the airflow from Asia continent. The ships intercepted with the air plumes with elevated radioactive materials, resulting in the enhancement of air dose rate measurements onboard the ships.

Earth System
Open Access  Science  Discussions
Data

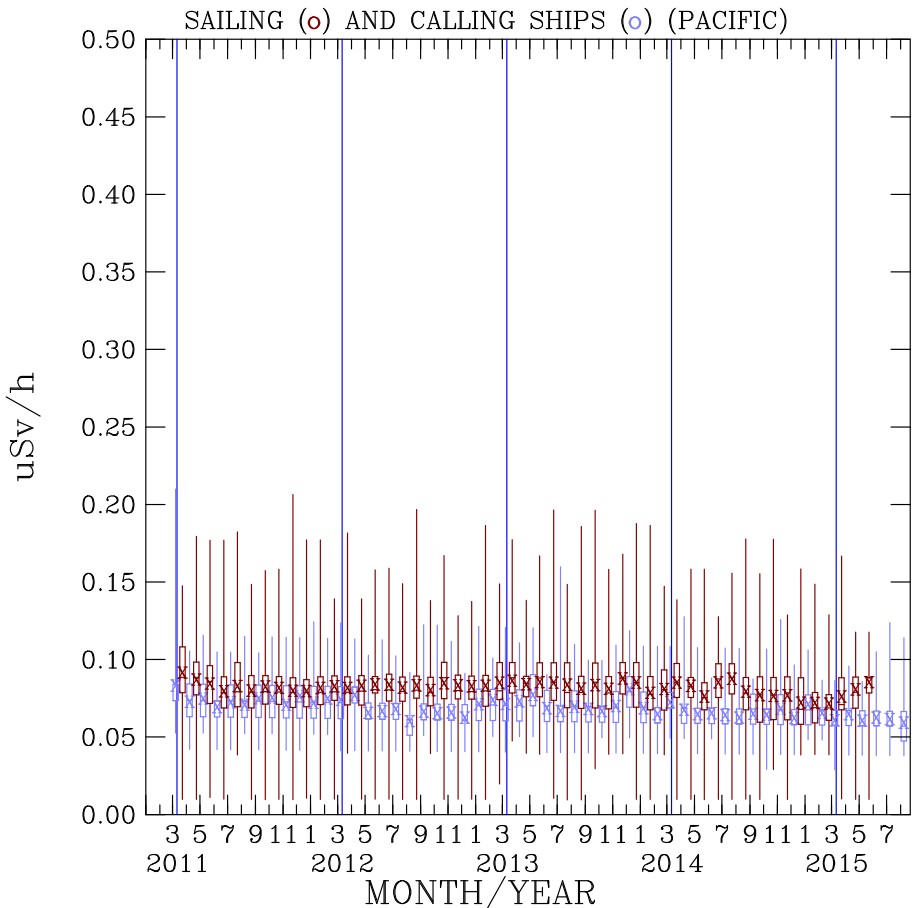

**Figure 10.** Comparison of radioactivity dose rates measured made on the Pacific sailing ships (brown colored) with those measured on the calling ships at the Tokyo Port (blue colored).

Fig. 10 compares Pacific sailing ship-based measurements with the Tokyo Port calling ships. The mean and maximum radioactive dose rates from the Pacific sailing ships are consistently higher than those measured on the Tokyo Port calling ships. These are important results. As the North Pacific atmosphere was directly downwind of Japan, the elevated air dose rates indicate the impact of radioactive materials being transported out of Japan to the North Pacific atmosphere. Impacts of elevated

air dose rates were higher over the North Pacific than on the calling ships over the ocean surface at the Tokyo Port.

On the other hand, low values of dose rates over the North Pacific are also consistently lower than those measured over the water surface of the Tokyo Port. Apparently, the cleaner air over the wide open Pacific contributed to the lowest measurements of the Pacific measurements than the Tokyo Port measurements. The large variabilities of dose rates over the North Pacific indicate the sensibility of the North Pacific to the upwind sources of air.



**3.5 Comparison with Other Published Measurements**

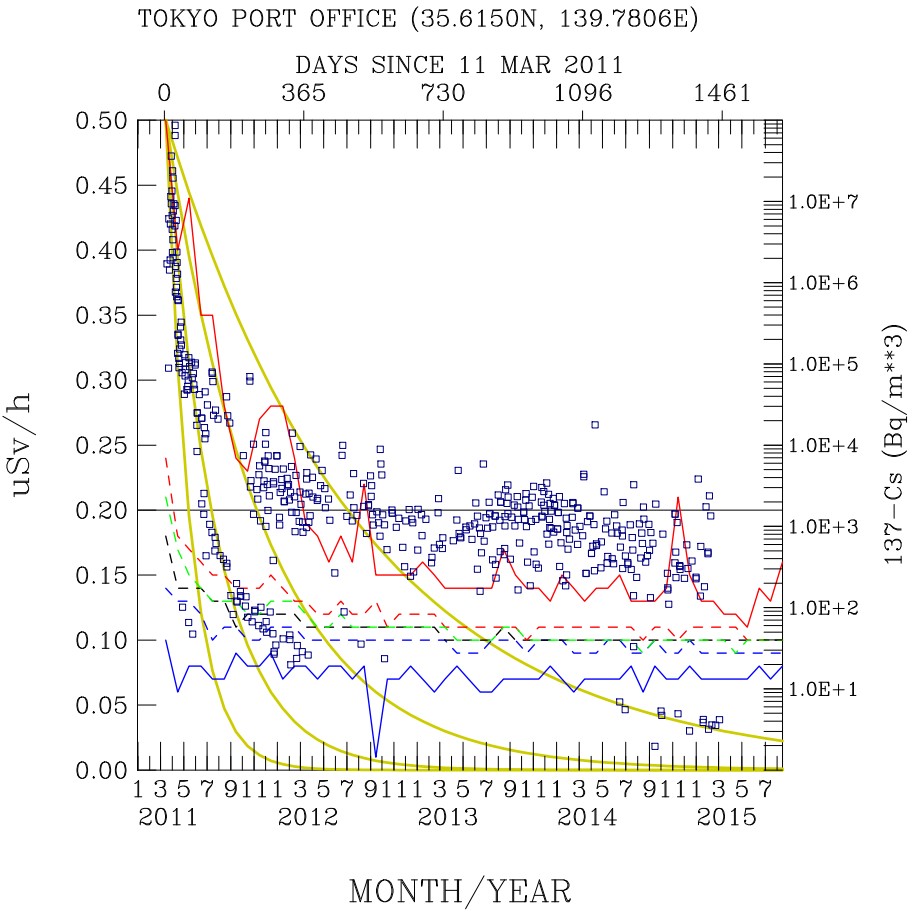

**Figure 11.** Comparisons of radioactivity measured made at the Tokyo Port with the radioactive nuclei $^{137}$Cs measured (blue squares, Bq $m^{-3}$) in a drainage by Aoyame et al. (2015). Red solid, red dashed, green dased, black dashed, blue dashed, and blue solid lines indicate maximum, the 75th, the 50th, mean, the 25th, and the minimum air dose rates measured at the Tokyo Port. For references, yellow curves show analytical distribution of time-series air dose rates calculated with a half-life of 45, 90, 180, and 360 days, respectively.

Fig. 11 compares the activities of 137 Cs in the surface water of canal units 5 and 6 of the Fukushima Daiichi site [see Fig. 2 of Aoyama et al., 2016] to radioactive dose rates measured at the attic of Tokyo Port office. The lowest activity of 137 Cs before the Fukushima accidents was around 1-2 Bq $m^{-3}$ [Aoyama et al., 2016]. The measurements at the two canal units of Fukushima nuclear power plant indicate a significant input of radioactive 137 Cs into the water. The initial values are 195 over 10 million Bq $m^{-3}$. It has taken about 2 years for this high levels of 137 Cs to decay to around thousands of Bq $m^{-3}$ exponentially. The pattern of exponential decay of monthly maximum levels of radioactive dose rates at the Tokyo Port office remarkably resembles the pattern of exponential decay of 137 Cs measured at the canal sites of the Fukushima nuclear power





plant. This comparison indicates that radioactive materials in Tokyo followed closely to that of the radioactive materials in the surface water of the canal of the Fukushima nuclear power plant.

The 75th, the 50th and the 25th percentiles, and mean of the dose rates also shows decay patterns but the reduction rates are less than those seen in the reduction rates of the monthly maximum levels of dose rates. Interestingly, the monthly lowest levels of dose rates were persistently close to 0.07 $\mu$Sv/h, except in September 2012. These lowest dose rates were not perturbed by the Fukushima accidents, indicating the background levels of radioactive dose rates over the Tokyo area.

Table 3 shows a list of published measurements of air dose rates made from sites in Japan after the Fukushima accident.
Elevated air dose rates of 94 $\mu$Sv/h was measured at the Daini Nuclear Power Plant (FNPP2), 15 km south of FNPP1 on 15 Mar 2011. Air dose rates at the Tsukuba site, located at 176 km south south west of the FNPP1, was measured at 0.2-0.4 $\mu$Sv/h during 20-21 March 2011. The measurements at the Tsukuba site are consistent with the measurements at the Tokyo Port reported in this work.

### 3.6   Temporal-Spatial Distribution of Radioactivity Levels over the North Pacific

Fig. 12a shows time-series measurements of 6-hourly maximum radiative dose rates over the northwestern Pacific regions (from $120°E$ to $180°E$, and from $20°N$ to $80°N$). The 6-hourly minimum, average, and all data are shown in Supplementary Material. The maximum levels of dose rates close to 0.18 $\mu$Sv were measured during April-July 2011. Transport of radioactive materials from the Fukushima and Japan areas after 11 Mar 2011 were measured over the northwestern Pacific.

Radiative dose rates had decreased in the subsequent months of August to December 2011. However, the dose rates went up
again in January 2012; in May-August and October 2012; in January, March, and June-September 2013; and in July-October 2014. Importantly, these are also the periods showing elevated 137 Cs measurements in the surface water of Fukushima nuclear power plant (see Fig. 11 and in Aoyama et al., 2016). Hence, These data indicate that sporadic waves of radioactive materials being deposited into the water and the air. The measurements were able to pick up signals from these unusual events.

Fig. 12b shows time-series measurements of 6-hourly maximum radioactive dose rates over the northeastern Pacific, further
downwind of the Fukushima and Japan areas. The monthly maximum dose rates show distinctively increased trends during the following twelve periods coinciding with the long-range of Asian pollutants to the northeastern Pacific : April-July 2011, August-November 2011, December 2011-January 2012; February-May 2012; June-August 2012; November-December 2012; January-May 2013; July 2013-January 2014; February-April 2014; June-September 2014; November-December 2014; February-March 2014. The gap months within these twelve periods of upward trends exhibit short periods of downward trends
in dose rates. The patterns of dose rates increases in a few months followed by a decline in the following months repeatedly occurs during April 2011-May 2015. These observed phenomena of up down in dose rates indicate the impact of the radioactive materials been transported from land areas to the northeastern Pacific. The months of increase in dose rates mostly occur from November to April, and from June to August. The November-April is months coincide with the active long-range transport of Asian dust and pollutant from Asian land areas downwind, passing Japan area, to the northeastern Pacific. This pattern
indicates long-range transport of Asian pollutants is an important mechanism responsible for the elevated levels of radiative



**Table 3.** List of Published Measurements of Radiation Dose Rates.

| Location | Distance to FNPP1 | Direction to FNPP1 | Dose Rates ($\mu$Sv/h) | Time | Reference |
|---|---|---|---|---|---|
| Minamisoma | 25 km | North North West | 20 | 12/03/2011 | Saunier et al. (2013) |
| Minamisoma | 25 km | Noth North West | 20 | 12/03/2011 | Tsuruta et al. (2014) |
| Minamisoma | 25 km | North North West | 20 | 20:00, 12/03/2011 | Korsakissok et al. (2013) |
| Tsukuriya Hokota City | 139 km | South | < 0.1 | 14/03/2011 | Saunier et al. (2013) |
| Koriyama | 59 km | West | 8 | 15/03/2011 | Saunier et al. (2013) |
| Aizuwakatmasu | 98 km | West | 2.5 | 15/03/2011 | Saunier et al. (2013) |
| Iitate | 39 km | North West | 45 | 15/03/2011 | Saunier et al. (2013) |
| Utsunomiya | 140 km | South | 1.3 | 15/03/2011 | Saunier et al. (2013) |
| Ishikawa | 126 km | South | 1.5 | 15/03/2011 | Saunier et al. (2013) |
| Minamiaizu | 114 km | West | 0.9 | 15/03/2011 | Saunier et al. (2013) |
| Shirakawa | 80 km | West | 7.5 | 15/03/2011 | Saunier et al. (2013) |
| Daini (FNPP2) | 15 km | South | 94 | 00:00, 15/03/2011 | Korsakissok et al. (2013) |
| Iwaki | 40 km | South South West | 23.7 | 01:00, 15/03/2011 | Korsakissok et al. (2013) |
| Tsukuriya Hokota City | 139 km | South | 4 | 07:00, 15/03/2011 | Saunier et al. (2013) |
| Kawauchi | 22 km | South West | 11.5 | 11:00, 15/03/2011 | Korsakissok et al. (2013) |
| Koriyama | 60 km | West | 6 | 14:00, 15/03/2011 | Korsakissok et al. (2013) |
| Iitate | 40 km | North West | 39.50 | 15:00, 15/03/2011 | Korsakissok et al. (2013) |
| Fukushima | 61 km | North West | 24 | 16:00, 15/03/2011 | Korsakissok et al. (2013) |
| Tsukuba | 176 km | South South West | 1 | 15/03/2011 | Tsuruta et al. (2014) |
| Momijiyama | 25 km | North West | 16 | 15 − 16/03/2011 | Tsuruta et al. (2014) |
| Tsukuriya Hokota City | 139 km | South | 1.5 | 06:00, 16/03/2011 | Saunier et al. (2013) |
| Yamamoto | 126 km | North | 1.6 | 16/03/2011 | Saunier et al. (2013) |
| Yamagata | 111 km | North | < 0.1 | 16/03/2011 | Saunier et al. (2013) |
| Minamisoma | 25 km | Noth North West | 6-8 | 18 − 21/03/2011 | Tsuruta et al. (2014) |
| Fukushima Tunnel | 61 km | North West | 0.4 | 19/03/2011 | Kubota et al. (2013) |
| Fukushima Tunnel | 61 km | North West | 1.0 | 20/03/2011 | Kubota et al. (2013) |
| Tsukuriya Hokota City | 139 km | South | 0.8 | 12:00, 20/03/2011 | Saunier et al. (2013) |
| Nihonmatsu City | 56 km | North West West | 5.8 | 20/03/2011 | Kubota et al. (2013) |
| Fukushima Tunnel | 61 km | North West | 2.0 | 21/03/2011 | Kubota et al. (2013) |
| Fukushima City | 61 km | North West | 2-9 | 19 − 22/03/2011 | Kubota et al. (2013) |
| Tohoku Expressway | 64 km | North West | 2-6 | 19 − 22/03/2011 | Kubota et al. (2013) |
| Banetsu Expressway | 64 km | West | 1.5-3 | 19 − 22/03/2011 | Kubota et al. (2013) |
| Samegawa Village | 64 km | South West | 1 | 19 − 22/03/2011 | Kubota et al. (2013) |
| Momijiyama | 25 km | North West | 7-9 | 20 − 21/03/2011 | Tsuruta et al. (2014) |
| Tsukuba | 176 km | South South West | 0.2-0.4 | 20 − 21/03/2011 | Tsuruta et al. (2014) |
| Tsukuriya Hokota City | 139 km | South | 2.2 | 06:00, 21/03/2011 | Saunier et al. (2013) |
| Nihonmatsu City | 56 km | North West West | 0.17 | 09/2011 | Fujimura et al. (2017) |
| Nihonmatsu City | 56 km | North West West | 0.17 | 2012 | Fujimura et al. (2017) |
| Nihonmatsu City | 56 km | North West West | 0.11 | 2013 | Fujimura et al. (2017) |
| Nihonmatsu City | 56 km | North West West | 0.07 | 03 − 05/2014 | Fujimura et al. (2017) |

dose rates measured over the northeastern Pacific. As such, the ship-based dose rate measurements reveal for the first time that the northeastern Pacific are particularly susceptible to the long-range transport of radioactive materials from Asian land areas.

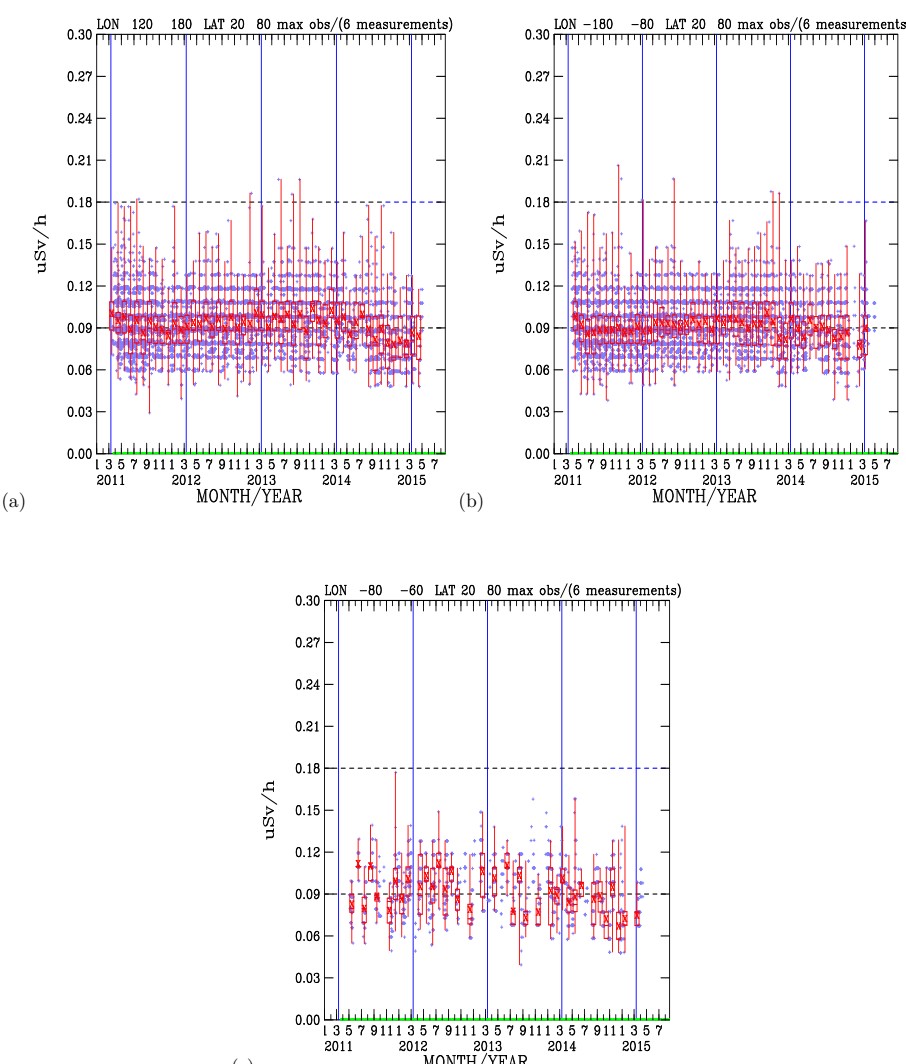

**Figure 12.** Time-series maximum air dose rates over the North Pacific and the Northwestern Atlantic atmosphere. (a) $120°E - 180°E$; (b) $180°W - 80°W$; (c) $80°W - 60°W$. Blue dots indicate all data; red lines are monthly statistics.

Fig. 12c shows time-series radioactive measurements further downwind of the northeastern Pacific, from $80°W$ to $60°W$ and over the northwestern Atlantic. The monthly mean of the 6-hourly maximum radiative dose rates show increase trends during May-June, July-August, and November-December in 2011. This is a stark contrast for the measurements made over the Pacific areas, which show decrease trends in radioactive dose rates during March-April 2011 over the northwestern Pacific (Fig. 12a), and during April-June 2011 over the northeastern Pacific (Fig. 12b). The direct impacts of the radioactive materials prevailed over the northwestern Pacific areas (March-April), then to the northeastern Pacific (Apri-June), and finally to reach the northwestern Atlantic (May-June).

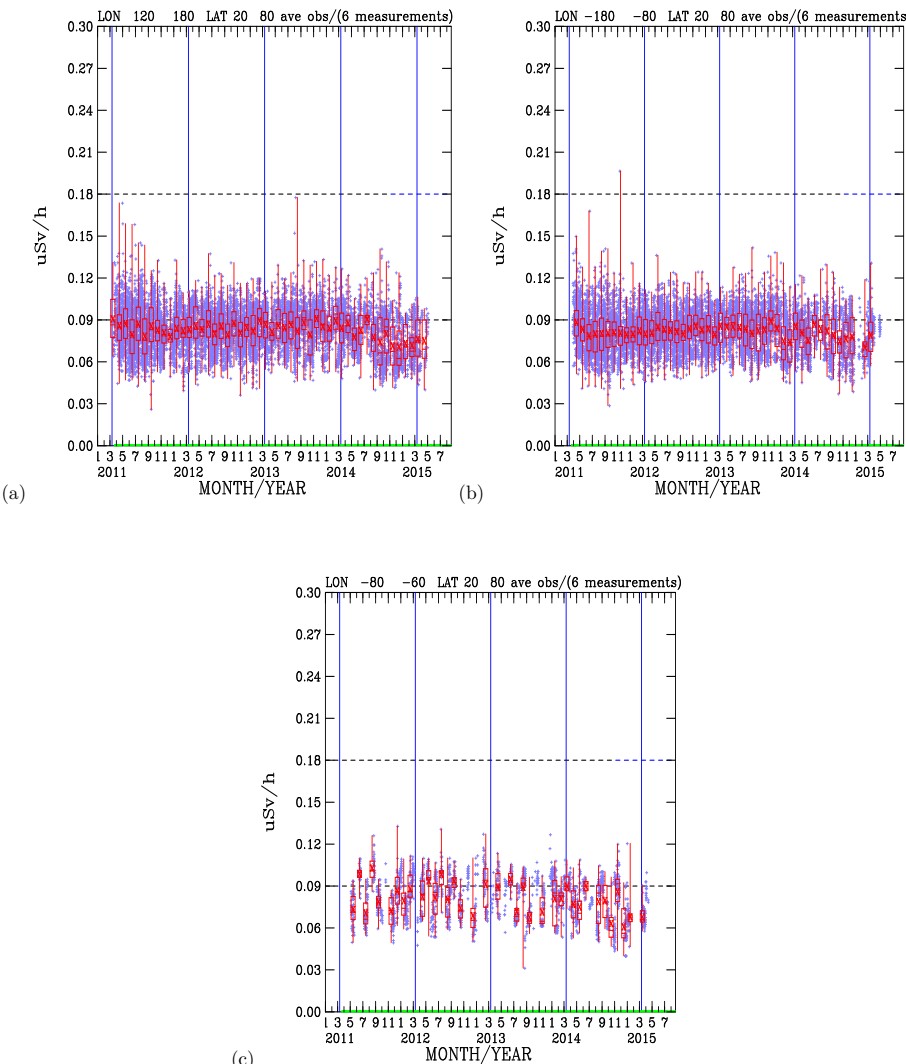

**Figure 13.** Time-series averaged air dose rates over the North Pacific and the Northwestern Atlantic atmosphere. (a) $120°E - 180°E$; (b) $180°W - 80°W$; (c) $80°W - 60°W$. Blue dots indicate all data; red lines are monthly statistics.

The impact of radioactive materials over the northwestern Pacific can also be seen when the average of 6-hourly measurements are plotted against time (Fig. 13a). The distinctive levels of maximum radioactive dose rates close to 0.18 $\mu$Sv/h in April, and then gradually decay to 0.11 $\mu$Sv/h in December 2011 indicates the presence of radioactive materials. Except for during August 2013, the maximum levels of dose rates from April to August 2011 are the highest values of all period. Over the northeastern Pacific, elevated radioactive dose rates higher than 0.15 $\mu$Sv/h were observed in April, June, and November of

2011 (Fig. 13b). For the northwestern Atlantic, the highest monthly mean dose rates were observed in August 2011 (Fig. 13c).


The time distribution of these elevated dose rates all indicates the impact of radioactive materials over the North Pacific and the northwestern Atlantic following the Fukushima disaster.

### 3.7 Spatial Distribution of Radioactivity Levels over the North Pacific

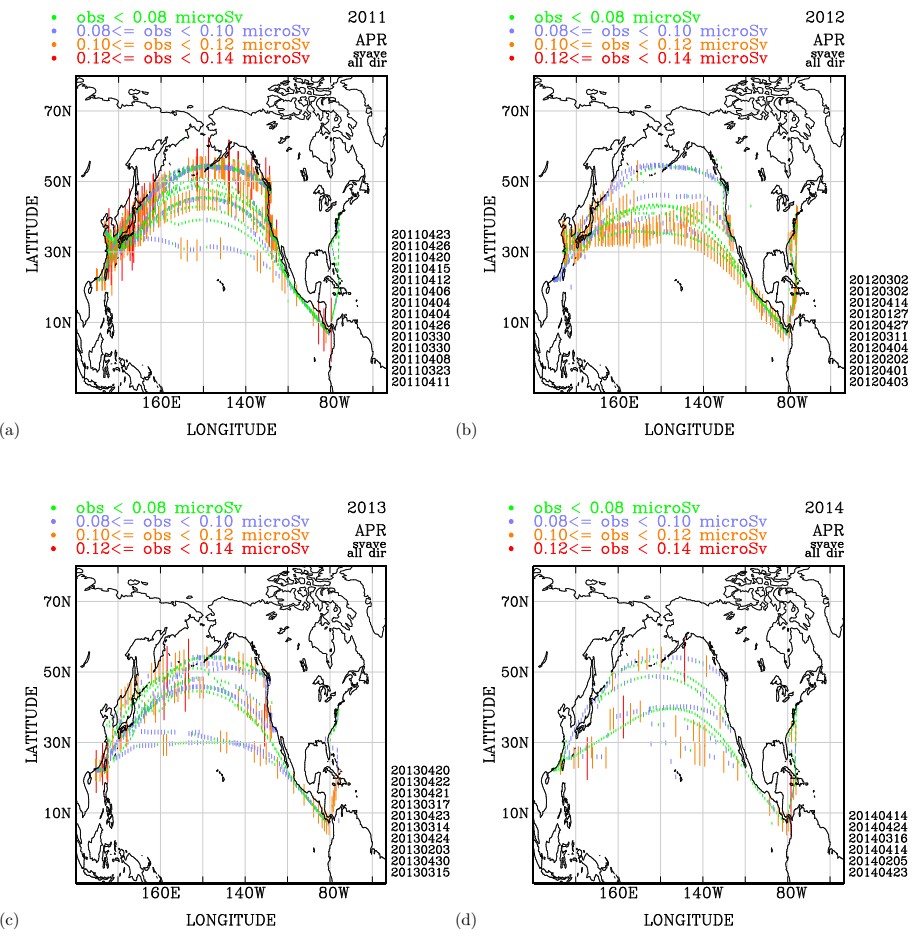

**Figure 14.** Measurements of radioactivity dose rates over the North Pacific during 2011-2015. (a) April 2011; (b) April 2012; (c) April 2013; (d) April 2014.

The continuous operations of the radioactive dose rate sensors onboard EMC ships from 2011 to 2015 had collected dose
rate data over the North Pacific shipping routes. These data enable us to understand dose rate distribution over the North Pacific after the Fukushima nuclear accidents. Fig. 14 compares the spatial distribution of radioactive dose rates measured in April of 2011, 2012, 2013, 2014, and 2015. Elevated dose rates mostly occur over the North Pacific in April 2011 than in the same month of subsequent years.




## 4   Summary

In this work, we identify three key aspects of the impacts of the radioactive materials following the Fukushima accident: The altitude effect, the land surface effect, and the transported effect. Comparisons of air dose rates measured at various altitudes from the ground surface indicate the altitude affect from the transported and deposited radioactive materials on the ground. Comparisons of measurements of air dose rates made over the land surface at the Tokyo Port and the measurements made over the ocean surface on the calling ship berthed at the Tokyo Port indicates the impacts of deposited radioactive materials

on the the land surface on measured air dose rates. Comparisons of measurements of air dose rates made at the Tokyo Port and the Pacific sailing container ships over the North Pacific atmosphere indicates waves of downwind transport of air riched in radioactive materials been transported downwind to the North Pacific atmosphere. The data of air dose rates over the North Pacific atmosphere show the eastward transport of materials containing radioactivity. The elevated observations in 2013 clearly indicate the active transport of radioactive materials. Eastward transport of radioactive materials been observed after 11 Mar

2011. Monitoring data show the export of radioactive materials to the Pacific atmosphere also occurred in 2012, 2013, 2014, and 2015.

With air dose rates reported in this work, and also data reported in previously published results, we will be able to use atmospheric models to test and verify the amount of the radioactive materials been emitted into the atmosphere following the Fukushima accident occurred on 11 March 2011 (Saunier et al., 2013.).

## 5   Data availability

All raw data files reported in this work are available for public downlad at https://doi.org/10.6084/m9.figshare.975769 (Wang et al., 2019), and the calibrated data files are available at http://doi.org/10.5281/zenodo.3468896 (Wang, 2019). The data we submitted is reachable with one click (without the need for entering login and password), and a second click to download the data, consistent with the two-click access principle for data published in ESSD (Carlson and Oda, 2018).

*Author contributions.*

KYW and NH designed the experiments and KYW carried them out. KYW, PN,and HC performed the analysis. KYW and PN prepared the manuscript with contributions from all co-authors.

*Competing interests.*

The authors declair that they have no conflict of interest.



*Acknowledgements.* We thank Evergreen Marine Corporation (EMC) for participating in the PGGM/IAGOS project. We are very grateful to the Taiwan Ministry of Science and Technology and Environmental Protection Administration for funding the PGGM project; and the European Commision for funding the IAGOS project.



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
