# Peer review of "Measurements of atmospheric radioactivity dose rates over the North Pacific after the Fukushima Daiichi nuclear power plant accident during the period March 2011 - March 2015"

_Earth System Science Data, 2019_

## Referee Comment (RC1) · Anonymous Referee #1 · 24 Oct 2019

Review ESSD-2019-156, atmospheric radioactivity dose rates over the North Pacific

Data download easily and cleanly from Zenodo link. Colletion of g01 to g14 folders each holding multiple .csv files. I believe one could find and re-assemble data to reproduce, e.g., one of the panels of Figure 14, but that effort would require significant time and work on the part of users. More seriously, as itemized below, this reviewer can not verify nor trust the author's descriptions in the text. I urge editors to reject and authors to revise and perhaps resubmit.

Specific comments:

P2L4: FNPP disaster happened 11 March 2011, not 11 April 2011?

P4L49: Again, this date seems wrong? FNPP disaster happened 11 March 2011, not 11 April 2011?

P5L62: "are openly and openly available"? You mean 'freely and openly available' or simply 'openly available'?

P5L63,64: "data will aid the validation of atmospheric models for modeling atmospheric
64 dispersions of radioactive materials from the nuclear power plant". Confusing and redundant. I think you mean 'data will assist evaluation and validation of models of atmospheric dispersions of radioactive materials from the nuclear power plant'?

P5L81: These dates imply that monitoring started very soon after the earthquake, tsunami and disruption at FNPP1? E.g. 11 March 2011 or very soon thereafter. Thus earlier dates of earthquake occurring on 11 April 2011 cannot be correct? Table 1 confirms start dates of 03/2011.

P6L85-89: Give us the manufacturer's specs for these sensors! Accuracy? Precision? Normal operating temperatures? Software version if any?

P6L102: What does the parenthetical phrase "(1 July 1996)" indicate here? The dates of the most recent confirmation of 137-Cs concentrations/activities? This date occurs in neither Figure 1 nor Table 2? Please clarify?

P7L110: "are consistently within 90designated dose rates". What does this mean. Can the authors provide a standard uncertainty, e.g. $\pm$ sd or $\pm$ 95%, or percentiles as used in the Figures?

P7L113: "(within 5(packages G03, G05, G06, and G07)."? Something missing here? From Figure 1, data from sensor G09, also a G10 sensor, look as good as data from G03, G05, G06, and G07?

P7L127: "taken at three locations". The G01 sensor moved small distances between port locations with time? Each time? Figure nor Table give the reader any information about how the authors sampled these three locations. Later, we learn 00UT and 06UT each day, but, again, sensor moved among three locations or locations changed with time? If the former, authors should have good error/uncertainty statistics. If the latter, readers needs to know exactly when the locations changed? Given data files and Figures 4, 5 and 6, it appears that all three locations were sampled each day. Authors need to clarify!

P8L141: "50 cm and 10 cm above the ground at the attic". 50 cm and 10 cm not very high. Confusion here? These measurements taken each day? G01 data folder shows three files: a4, outdoor and vessel. A4 = Gate? Outdoor = ATTIC? Vessel = over the water at the port? Only _outdoor file shows obs related to height above ground? No height-related differences?

P8L149 and following: I can not find or confirm these data. First, the authors mean the highest individual data (from twice daily measurements), not the "highest monthly radioactive dose". Second, in the _outdoor data file, I find maximum values of 0.50 and 0.49 microSv/h. Nothing to show how the authors got 0.40 or 0.45. I find no values of 0.45 in that entire table?

We get no information of how the authors processed or filtered these data. Did they average all directional measurements, e.g. N, E, W, S, together? Did they add UP and DOWN measurements, to take an average of 6 separate data points? Did they give preference to 50 cm vs 10 cm? Did they even analyze 50 cm vs 10 cm or UP vs DOWN? All the data files hold this basic structure (time, N, E, W, S, UP, DOWN), with additional lat lon data from the ships, but reader gets no idea about averaging, filtering, etc.

P8L151 and following: "0.18 Sv/h in March 2012" I can not find nor reproduce that value. I calculate an average of 0.12 for March 2012 with a max of 0.14 (22 data points). What gives? "0.15 Sv/h in March 2013"? I can not find nor reproduce that value (I calculate average 0.10, max 0.12, for 20 data points), nor any of the following values listed in this paragraph.

Figure 4: vertical blue lines indicate years after FNPP accident?

Figure 5 (and many other figures): The dose meter data in all these figures shows discrete rather than continuous values. One might suspect bit noise but we get no information about data resolution. We do see the impact of the start-to-end calibration because these discrete values clearly decline in magnitude over time while still retaining their discrete distributions?

At this point I gave up. Until the text matches data, or data matches the text, or the authors give better clearer instructions to reviewers and potential users, I see no point in proceeding.

c

---

## Author Comment (AC1) · 3 Nov 2019

Review ESSD-2019-156, atmospheric radioactivity dose rates over the North Pacific

Data download easily and cleanly from Zenodo link. Colletion of g01 to g14 folders each holding multiple .csv files. I believe one could find and re-assemble data to reproduce, e.g., one of the panels of Figure 14, but that effort would require significant time and work on the part of users. More seriously, as itemized below, this reviewer can not verify nor trust the author's descriptions in the text. I urge editors to reject and authors

to revise and perhaps resubmit.

Reply. We are very grateful indeed to the reviewer for very detailed comments. Detailed replies to each comment are shown in the following.

Specific comments:

P2L4: FNPP disaster happened 11 March 2011, not 11 April 2011?

Reply. We have made correction to the date of the FNPP disaster. It was on 11 March 2011.

P4L49: Again, this date seems wrong? FNPP disaster happened 11 March 2011, not 11 April 2011?

Reply. We have corrected the date to 11 March 2011.

P5L62: "are openly and openly available"? You mean 'freely and openly available' or simply 'openly available'?

Reply. We have rephrased to "openly available".

P5L63,64: "data will aid the validation of atmospheric models for modeling atmospheric dispersions of radioactive materials from the nuclear power plant". Confusing and redundant. I think you mean 'data will assist evaluation and validation of models of atmospheric dispersions of radioactive materials from the nuclear power plant'?

Reply. We have rephrased the description as suggested by the reviewer.

P5L81: These dates imply that monitoring started very soon after the earthquake, tsunami and disruption at FNPP1? E.g. 11 March 2011 or very soon thereafter. Thus earlier dates of earthquake occurring on 11 April 2011 cannot be correct? Table 1 confirms start dates of 03/2011.

Reply. We have corrected the date to 11 March 2011.

P6L85-89: Give us the manufacturer's specs for these sensors! Accuracy? Precision?

Normal operating temperatures? Software version if any?

Reply. The sensors have an accuracy of $\pm 3\%$ (please attached two pdf files for the original calibration certificates from the makers). Our calibrations of sensors (see Figure 1) show that the sensors have the precisions of 2% (50th percentile) and 5% (75th percentile) of deviations to the designated dose rates. The sensors have an operating temperature range of -20oC to 50oC (SAVER, 2016; RedEye B20, 2017). We have included above description in the end of the first paragraph of section 2.2.

P6L102: What does the parenthetical phrase "(1 July 1996)" indicate here? The dates of the most recent confirmation of 137-Cs concentrations/activities? This date occurs in neither Figure 1 nor Table 2? Please clarify?

Reply. This is the date of the 137-Cs source obtained in calibrating the sensors used in this work. We have added this description after the date.

P7L110: "are consistently within 90designated dose rates". What does this mean. Can the authors provide a standard uncertainty, e.g. + sd or + 95%, or percentiles as used in the Figures?

Reply. We have followed reviewer's suggestion to rephrase this descript as: "are consistently at 1% (50th percentile) and 4% (75th percentile) of deviations to the designated dose rates".

P7L113: "(within 5(packages G03, G05, G06, and G07)."? Something missing here? From Figure 1, data from sensor G09, also a G10 sensor, look as good as data from G03, G05, G06, and G07?

Reply. We are very grateful indeed to the reviewer for pointing out this incomplete description. The error was caused by a % symbol after 5% used in the original latex file, which was converted from a word file. We have fixed and rephrased this description following reviewer's comment.

P7L127: "taken at three locations". The G01 sensor moved small distances between

port locations with time? Each time? Figure nor Table give the reader any information about how the authors sampled these three locations. Later, we learn 00UT and 06UT each day, but, again, sensor moved among three locations or locations changed with time? If the former, authors should have good error/uncertainty statistics. If the latter, readers needs to know exactly when the locations changed? Given data files and Figures 4, 5 and 6, it appears that all three locations were sampled each day. Authors need to clarify!

Reply. Yes, all three locations were sampled each day by the same G01 sensor. We have included this description in the revised manuscript.

P8L141: "50 cm and 10 cm above the ground at the attic". 50 cm and 10 cm not very high. Confusion here? These measurements taken each day? G01 data folder shows three files: a4, outdoor and vessel. A4 = Gate? Outdoor = ATTIC? Vessel = over the water at the port? Only _outdoor file shows obs related to height above ground? No height-related differences?

Reply. We have included following description to increase the clarity of the data: "In the raw data files (Wang et al., 2019) and calibrated data files (Wang, 2019), there are three files in the G01 data folder: a4, outdoor and vessel. A4 represents measurements made at gate. Outdoor represents measurements at office attic. Vessel represents measurements over the the water at the port. Only the outdoor file shows measurements related to height about the ground. The height-related measurements are shown in the last three columns of data (DOWN, 50 cm, and 10 cm) of the outdoor files."

P8L149 and following: I can not find or confirm these data. First, the authors mean the highest individual data (from twice daily measurements), not the "highest monthly radioactive dose". Second, in the _outdoor data file, I find maximum values of 0.50 and 0.49 microSv/h. Nothing to show how the authors got 0.40 or 0.45. I find no values of 0.45 in that entire table?

[Figure]

Reply. We have revised the description as suggested by the reviewer: "In the first two months after 11 March 2011, the highest individual radioactive dose rates measured at Tokyo Port office is at 0.40 $\mu$Sv/h (at 0600 UT on 23 Mar 2011 of 20110322_20150902_tokyo_outdoor.txt; Wang et al., 2019). In the following discussions, the highest individual radioactive dose rates measured twice daily in a month is referred to as the highest monthly dose rates. The highest monthly dose rates gradually drop with time."

The calibrated data measured at 0600 UT on 23 Mar 2011 is 0.42 $\mu$Sv/h in the 20110322_20150902_tokyo_outdoor.csv (Wang, 2019).

P8L149 We get no information of how the authors processed or filtered these data. Did they average all directional measurements, e.g. N, E, W, S, together? Did they add UP and DOWN measurements, to take an average of 6 separate data points? Did they give preference to 50 cm vs 10 cm? Did they even analyze 50 cm vs 10 cm or UP vs DOWN? All the data files hold this basic structure (time, N, E, W, S, UP, DOWN), with additional lat lon data from the ships, but reader gets no idea about averaging, filtering, etc.

Reply. We have rephrased following description in the beginning of section 3.1: "Fig. 4 shows time-series measurements of radioactive dose rates at the attic of Tokyo Port office from March 2011 to September 2015. The green dots show all measurement data (from all six directions and at two heights of 10 cm and 50 cm above the ground)."

No averaging and filtering processes were used in this work. We presented all measurement data.

P8L151 and following: "0.18 Sv/h in March 2012" I can not find nor reproduce that value. I calculate an average of 0.12 for March 2012 with a max of 0.14 (22 data points). What gives? "0.15 Sv/h in March 2013"? I can not find nor reproduce that value (I calculate average 0.10, max 0.12, for 20 data points), nor any of the following values listed in this paragraph.

Reply. The dose rate of 0.18 uSv/h appear in raw data (Wang et al., 2019). We have rephrased the description as following: "It was 0.18 $\mu$Sv/h in March 2012 (at 2330 UT on 13 March 2012, in raw data of Wang et al. (2019)), one year after the disaster."

Figure 4: vertical blue lines indicate years after FNPP accident?

Reply. We have rephrased the figure caption for Figure 4 as following: "The vertical red line indicates the maximum and minimum air dose rates measured in a month. Vertical blue lines indicate 11 March of 2011, 2012, 2013, 2014, and 2015, respectivey."

Q. Figure 5 (and many other figures): The dose meter data in all these figures shows discrete rather than continuous values. One might suspect bit noise but we get no information about data resolution. We do see the impact of the start-to-end calibration because these discrete values clearly decline in magnitude over time while still retaining their discrete distributions?

Reply. As indicated in a previous reply, the precisions of the sensors are within 2% (50th percentile) and 5% (75th percentile) of the deviations from the 4 designated dose rates. Hence, the sensors are quite accurate and all the raw data (Wang et al., 2019) were calibrated and presented in this work (Wang, 2019). The gradual decline in magnitude of dose rates over time are consistent with the decline of radioactivity with time over the source region and land area, as shown in Figures 4 and 11.

Q. At this point I gave up. Until the text matches data, or data matches the text, or the authors give better clearer instructions to reviewers and potential users, I see no point in proceeding.

Reply. We are very grateful to the reviewer for the very insightful comments. The raw data (Wang et al., 2019) and the calibrated data in csv format (Wang, 2019) were uploaded. It was the original raw data values that were referred in the text of the submitted manuscript.

Please also note the supplement to this comment:
https://www.earth-syst-sci-data-discuss.net/essd-2019-156/essd-2019-156-AC1-supplement.zip

---

## Editor Comment (EC1) · David Carlson (Editor) · 6 Nov 2019

Please can we start over on this manuscript and data set? Having seen reviewer comments and read the manuscript myself, ESSD could not accept this initial product. Wrong dates, technical errors, descriptions do not match data, absence of uncertainty information, etc. It seems obvious to this editor that neither author nor co-authors did due diligence, e.g. proofreading, before submittal. Now author has evidently further contaminated the process by changing or adding data. Subsequent reviewer(s) will not see the same data as the first review?

[Figure]

Please withdraw the current manuscript. Check all data and data errors. Prepare a clean data set, preferably of calibrated values double-checked and with all available uncertainties explicitly documented. Register data under a new DOI. Resubmit a revised manuscript, after careful checking by all authors.

Taking these steps will not slow down ESSD review and evaluation processes. Those processes might actually go faster via these steps.

---

## Author Comment (AC2) · 6 Nov 2019

We are very grateful indeed to Editor's comments and suggestions. We want to follow Editor's suggestion to withdraw this version of manuscript so as to restart again. The revised manuscript will include a new data DOI for the calibrated data, and only the calibrated data be reported in the revised manuscript. All the data described in the revised manuscript to be carefully checked for consistency with the calibrated data uploaded. We are also very grateful again to the reviewer for pointing out locations in the first manuscript that require detailed and careful checkings. Thank you very much

indeed.